# Lower Extremity Flexibility Profile in Basketball Players: Gender Differences and Injury Risk Identification

**DOI:** 10.3390/ijerph182211956

**Published:** 2021-11-14

**Authors:** Antonio Cejudo

**Affiliations:** Department of Physical Activity and Sport, Faculty of Sport Sciences, Regional Campus of International Excellence “Campus Mare Nostrum”, University of Murcia, 30100 Murcia, Spain; antonio.cejudo@um.es; Tel.: +34-868-888-430

**Keywords:** team sports, sports risk, limited range of motion, asymmetric range of motion, muscle extensibility, optimal cut-off values, stretching

## Abstract

Analysis of the flexibility profile of basketball players (BPs) can reveal differences in range of motion (ROM) by gender and also identify those players who are at higher risk for sports injuries. A descriptive observational study was conducted to determine the lower extremity flexibility profile of sixty-four basketball players and gender differences to identify players at higher risk of injury due to limited and asymmetric ROM in one or more movements. Participants: Sixty-four (33 male and 31 female) competitive athletes from the national leagues of the Spanish basketball league system participated in the present study (power of sample size ≥0.99). The eight passive ROM tests of the hip, knee and ankle were assessed using the ROM-SPORT battery. Each player completed a questionnaire on age, basic anthropometric data, dominant extremities, and training and sport-related variables. The lower extremity flexibility profile was established at 15° and 10° hip extension (HE), 39° and 38° ankle dorsiflexion with knee extended (ADF-KE), 40° and 39° ankle dorsiflexion with knee flexed (ADF-KF), 43° and 43° hip abduction (HAB), 75° and 61° hip abduction with the hip flexed (HAB-HF), 78° and 83° hip flexion with the knee extended (HF-KE), 134° and 120° knee flexion (KF), and 145° and 144° hip flexion (HF) by male and female basketball players, respectively. Sex differences in HE, HAB-HF, and KF were observed in BPs (*p* ≤ 0.01; Hedges’ g ≥ 1.04). Players reported limited ROM in ADF-KF, HE, HAB-HF, HF-KE, and KF; and asymmetric ROM mainly in HE, ADF-KE, KF, ADF-KF, and HF-KE. In conclusion, this study provides gender-specific lower extremity flexibility profile scores in BPs that can help athletic trainers and athletic and conditioning trainers to identify those players who are at higher risk of injury due to abnormal ROM scores.

## 1. Introduction

Muscle extensibility is defined as the ability of some muscle tissue components to deform by increasing their length under the influence of an external and internal torque [1,2]. Changes in muscle extensibility are reflected in changes in joint angle when a standardised torque is applied (e.g., during a stretching program) [2,3,4]. Primarily muscle extensibility [1,5] and additionally other joint tissues are indirectly quantified by range of motion (ROM). In the field of sports and health sciences, the main objectives in measuring ROM are: (1) to identify flexibility-related risk factors such as muscle tightness and asymmetries for athletic performance [6] and sports injuries [7,8,9], and (2) to evaluate the physical-technical training process and rehabilitation of sports injuries [10,11]. For this purpose, sports and health professionals need the reference values for ROM and muscle extensibility during sports.

The method of measuring ROM has been established internationally by institutions such as the American Academy of Orthopaedic Surgeons [12] and the American Medical Association [13]. The average reference values for healthy joints in the general population have been published by both institutions. To set goals for flexibility training, sports and health professionals need reference values for athletes with or without pathology. When assessing ROM values in sports, differences in flexibility values have been found between different sports [14]. In addition, flexibility has been shown to depend on age [15,16], gender [14,17], lateral dominance due to muscle asymmetry [18,19], player position [20,21], and tier of competition [16,22]. It has been observed that an athlete who has specific ROM values for each technical movement of the sport has an optimal ROM that allows him to improve his physical and technical performance [23,24]. Several studies have observed that physical and technical performance in sports (sprinting, jumping, agility, kicking, balance) decreases when the technical movement is limited by muscle tightness [25,26] or non-optimal ROM [24]. Therefore, the success of the athlete requires specific or functional ROM values [24,27] that must be accompanied by optimal values in other sports performance parameters [28].

Knowing the reference values that define the lower extremity flexibility profile for their sport is of great interest to athletic trainers and athletic and conditioning trainers [15,24]. The mean ROM values of the major lower extremity movements in ascending or descending order for a group of athletes are defined by Cejudo as the lower extremity flexibility profile for that sport [15,24]. Lower extremity flexibility profile values are determined by performing tests that integrate the standardised ROM-SPORT battery for a group of athletes [15,24,29]. Previously published studies have determined the lower extremity flexibility profile by applying the ROM-SPORT battery to handball [30], soccer [15,31], futsal [17,32], and inline hockey players [18,29]. In addition, the ROM-SPORT battery predicted the risk of low back pain using the hip flexion test with the knee extended in basketball and soccer players [33], the external and internal rotation test in inline hockey players [34] and hip adduction and knee flexion in equestrian athletes [23].

At the same time, limited ROM due to muscle tightness [8,23,28,34,35] and muscle asymmetry or inter-limb ROM asymmetry [8,36,37] have been associated with sports injuries in several prospective cohort studies, and thresholds have even been established to distinguish athletes at higher risk for sports injuries. When sports and health professionals assess the ROM of their athletes, the results can be compared to the reference values (optimal cut-off values for predicting injury risk) of their sport to identify athletes at high risk of injury. As sports injuries consist of multiple components, the same injury risk identification strategy should be used for other risk factors such as body mass index, muscle strength, power, balance or coordination.

To our knowledge, previous studies have not identified lower extremity-based specificity analysis ROM for basketball players (BPs). Accordingly, the aim of the current study was twofold: (1) to define the lower extremity flexibility profile of sixty-four BPs along with gender differences, and (2) to identify players at higher risk of injury due to limited ROM or inter-limb ROM asymmetry in one or more movements.

## 2. Materials and Methods

This investigation (ID: 1702/2017) was approved by the Institutional Ethics Committee of the College of Murcia (Spain) and complied with the World Medical Association Code of Ethics, Declaration of Helsinki.

### 2.1. Design of Study

Seventy-three BPs from four basketball teams were initially recruited for this descriptive observational study. The players played in the second (36 female players) and fourth (37 male players) levels of the Spanish basketball league. In order to recruit the BPs to participate in our study, the researchers contacted the coaches one month before the assessment session. One week prior to the evaluation, an investigator met with the technical staff and BPs to explain the objectives of the study. This was also used to familiarize the BPs with the ROM tests. Familiarization with the testing procedures took place during the BPs’ first visit to the sports center during peak competition time. During the second visit, each player signed an informed consent form and completed the questionnaire that included demographic, anthropometric, and basketball training data. In addition, during the competition period, an assessment of essential lower limb movements was performed using the eight ROM tests from the ROM-SPORT battery [38].

Players were instructed not to engage in vigorous physical exertion 48 h prior to the study and were informed that they could withdraw from the study at any time. BPs performed a dynamic warm-up of approximately 20 min, following recommendations from a previous study [38]. All ROM measurements were performed in a private sports medicine room under standard 25 °C conditions. BPs were measured barefoot and in their normal sportswear. None of the BPs showed any musculoskeletal complaints or injuries during the study. Measurements were taken simultaneously by two experienced musculoskeletal assessment experts. The order of testing was randomized using the software at http://www.randomizer.org to avoid bias that might occur if the measurements were taken in a particular order. The average of the two measurements was used for further statistical analysis. A third measurement was taken if the investigators found a variation of more than 5% between these two measurements of any test to identify the error measurement. There was a pause of more than one minute between repetitions and two minutes between each test. The data were then analyzed to confirm or reject the null hypothesis.

Later, the lower extremity flexibility profile was determined and gender differences between BPs were analyzed. To complete the comprehensive analysis, each player’s ROM values were compared to reference values (optimal cut-off values for predicting injury risk) to identify players who were ROM limited by muscle tightness and inter-limb ROM asymmetry.

### 2.2. Basketball Players Sample

Sixty-four BPs aged between 17 and 31 years (average of 22.6 ± 2.9 years) participated as volunteers in this study and had an average of 12 years (12.4 ± 3.9 years) of basketball experience (Table 1). Thirteen BPs played in the fourth-tier grade of the Spanish men’s basketball league and eleven played in the second-tier grade of the Spanish women’s basketball league. None of the BPs had suffered orthopedic injuries to the lower extremities or lower back in the previous 14 days (e.g., sprained ankle, muscle injury, tendinopathy, low back pain, etc.), which could affect ROM, anthropometric characteristics (Table 1), and lower limb assessment ROM (hip, knee, and ankle). Players with delayed onset muscle soreness were excluded from this study.

The power of the sample size for this study was analyzed as described in the Statistical Analysis section.

### 2.3. Examiners

Two exercise and sports science graduates with at least 15 years of experience in musculoskeletal assessment conducted the data collection for this study. They were the same examiners and had the same competencies in assessing ROM. The principal examiner performed the ROM tests while the assistant examiner reviewed the compensatory movements and recorded the data [38]. In a double-blind study with 12 participants, the examiners reported good test-retest reliability for the ROM measurements (intraclass correlation coefficients equal to or greater than 0.94 for the ankle ROM and equal to or greater than 0.92 for the knee and hip ROM).

### 2.4. Interview Survey

The survey included questions about self-reported age, basic anthropometric data (height, body mass, and body mass index), dominant laterality (defined as the leg you would use for one-legged jumps to achieve maximum jump height), training, and sport-related variables (basketball experience, level of performance in basketball, hours of basketball training per week, and playing time per basketball competition). The data in the surveys were cross-checked with the basketball coach to increase objectivity. If the coach and player data did not match, club or federation records were used. The examiners evaluated the anthropometric measurements and verified the information provided in the questionnaires.

### 2.5. Method of Assessment

The maximum passive hip, one knee and ankle ROMs were assessed using the ROM-SPORT battery [38]. This battery provides a clear and concise description of the assessment procedure ROM such as the initial position of the participant, the assessment movement, the use of the measuring instruments (inclinometer, long arm metal goniometer, lumbar support), the competencies of both examiners, the maximum passive ROM, the conditions for the end of the test, the non-inclusion of compensatory movements in the measurement and the number of trials [38]. The hip extension test for the iliopsoas (HE), hip abduction with hip at 0° test for the adductors (HAB) and hip flexion at 90° test for the monoarticular adductors (HAB-HF), hip flexion with knee extended at 0° test for the hamstrings (HF-KE) and with knee flexed test for the gluteus maximus (HF-KF), knee flexion test for the quadriceps (KF), ankle dorsiflexion with knee flexed test for the soleus (ADF-KF) and knee extended at 0° test for the gastrocnemius (ADF-KE) are the tests that are part of the ROM-SPORT battery. These tests are designed so that the muscle at the end of the maximum is the protagonist of the joint movement ROM. The angle between the long axis of the mobilized lower limb (following its bisector) and the horizontal was assessed [38]. The ROM were measured using an ISOMED Unilevel inclinometer (ISOMED, Inc, Portland, OR, USA). The accuracy of the inclinometer (ISOMED, Inc, Portland, OR, USA) is two degrees. A rigid back support or Lumbosant© (Imucot Traumatología SL, Murcia, Spain) was used to place the pelvis in a neutral position (20°). The assistant examiner was responsible for the control of compensatory movements. Both the non-dominant and dominant lower limbs were assessed. The results of these measurements in ascending order determine the lower extremity flexibility profile, as described previously [38]. Previous studies have demonstrated the reliability [38] and validity [39,40] of the ROM assessment based on sports experience and biomechanical knowledge [12,13]. Previous studies by the research group have shown intra-evaluator variability of 4° to 7° for the ROM-SPORT battery [38].

### 2.6. Statistical Analysis of the Data

First, the assumptions about the normality of the data were tested using the Shapiro–Wilk normality test. Descriptive statistics with mean and standard deviation were calculated for all variables tested.

Second, a Student’s *t*-test for independent samples was conducted to determine possible gender differences in the demographic and athletic variables. A post hoc sample size calculation was performed using the software package G*Power 3.1.9.7. (Heinrich-Heine-Universität Düsseldorf, Düsseldorf, Germany) using a sample size of 33 female and 31 male basketball players, an alpha level of *p* < 0.05, effect size (Hedges’ g), and Student’s *t*-test for independent samples (Figure 2) [41].

Paired samples *t*-test was calculated to estimate inter-limb ROM asymmetry. To assess the effect size of the difference, Hedges’ g was calculated from the values of the paired samples *t*-tests and the effect was classified as no effect (g < 0.2), small: (g = 0.2 to 0.59), moderate (g = 0.6 to 1.19), large (g = 1.20 to 2.00), very large (g = 2.00 to 3.99), extremely large (g > 4.00) [42]. The lowest level of an important effect with practical application was found at moderate or higher than this effect size for gender demographic and athletic data and inter-limb ROM asymmetry [42].

A k-means cluster analysis was performed to classify participants between younger and older players. An independent-samples *t*-test was performed to assess differences between the ROM of younger and older players.

Cut-off values for risk factors according to previous prospective cohort studies were used to identify athletes at higher risk of injury with normal vs. inter-limb ROM asymmetry (6° for low ROM as HE, HAB-HF, HAB, and ADF-KE, ADF-KF) and 10° for the high ROM values as HF-KE, KF, and HF-KF) [8,19,34,36,37,43,44,45] and normal vs. limited ROM (13° in HE, 30° in ADF-KE, 37° in ADF-KF, 28° in HAB, 80° in HAB-HF, 88° in HF-KE, 132° in KF, and 135° in HF-KF) [11,12,13,46,47,48]. Finally, an independent-samples *t*-test was performed to assess differences between normal and limited ROM groups; the Hedges g effect size was also considered. Statistical analyses were performed using SPSS software (IBM Corp., Armonk, NY, USA), with a significance level of at least five percent.

## 3. Results

The initial sample consisted of 73 players. Nine players were withdrawn from the study by the researchers. The reasons were that the players had not completed the questionnaire correctly, had not signed the informed consent form, or had received long-term rehabilitation treatment (ankle sprain and low back pain) in the past six months, including flexibility training.

The variables (see Figure 2) identified in this study resulted in a sample size significance of 0.99 for HE ROM, 1.00 for HAB-HF ROM and 0.99 for KF ROM.

Inter-limb ROM asymmetry was found only for the quadriceps in KF ROM (*p* = 0.029); however, the effect size of the effect found between both sides of the body was small (Hedges’ g = −0.34). Figure 1 therefore shows the lower extremity flexibility profile (mean values of each ROM test) for the 33 male and 31 female BPs, respectively.

Differences were found between the group of young players (<23 years) and the group of older players (≤23 years) in the KF ROM (*p* = 0.40; Hedges’ g = 0.871). Figure 2 shows the gender differences in the lower extremity flexibility profile (mean values of each range of motion test). Differences were observed in HE ROM (iliopsoas (*p* = 0.01; Hedges’ g = 1.04)), HAB-HF ROM (monoarticular adductors (*p* = 0.00; Hedges’ g = 2.43)) and KF ROM (quadriceps (*p* = 0.00; Hedges’ g = 1.49)) with moderate, very large and large effect sizes, respectively. Male players showed higher values than female players in all three ROMs.

Of the 64 BPs, 42 (65.6%) of the players in HE (iliopsoas), 24 (37.5%) of the players in ADF-KF (soleus), 44 (68.8%) of the players in HAB-HF (monoarticular adductors), 39 (60.9%) of the players in HF-KE (hamstrings), 37 (57.8%) of the players for KF (quadriceps), and 18 (28.1%) of the players for HF ROM (gluteus maximus) showed limited ROM (Table 2).

Individual analysis of inter-limb ROM asymmetry revealed a slightly greater number of lower values in the dominant limb compared to the same movement in the contralateral limb (Table 3). Lower values were observed in the dominant limb in the HE (iliopsoas), and KF (quadriceps) ROMs, while in the non-dominant limb lower values were observed mainly in the ADF-KE ROM (gastrocnemius), ADF-KF ROM (soleus), and HF-KE ROM (hamstrings). The mean inter-limb difference in players classified as asymmetric group (values > 6° to 10° according to ROM test) was 8.2° and in normal asymmetric group (values ≤ 6° to 10° according to ROM test) was 3.7°.

## 4. Discussion

The present study is the first to describe the lower extremity flexibility profile in 64 BPs of both sexes using a standardised procedure. In general, the results suggest that the ROM-SPORT battery determines the first reference values of BPs competing in the national-tier leagues of the Spanish basketball league system. However, differences were observed between younger and older players in KF ROM, which is consistent with the results of previous studies [15,16]. The flexibility profile resulting from our study shows some gender differences (Figure 1). In general, it has always been hypothesised in the sports field that female players have higher values in the lower extremity flexibility profile than male players. However, after analyzing the ROM lower extremity flexibility profile values, our study completely rejects this original hypothesis. Male players show higher values in HE ROM (iliopsoas), HAB-HF ROM (monoarticular adductors) and KF ROM (quadriceps) than female BPs, which will contribute to better execution of the technical gesture and better athletic performance compared to female players [22,25,49]. Despite the fact that females have biological advantages that favour greater ROM such as smaller muscle volume, larger pelvic diameter, and lower centre of gravity [17,50], the results of the present study contradict the assumption that females are more flexible than males. A systematic review found that basketball demands and physiological data vary between male and female BPs at the same level of competition [51]. In this regard, it seems that the sport played influences the ROM for certain movements according to the gender of the player. The basic playing position of a basketball player is a partial squat with significant bilateral hip abduction. This position requires adequate extensibility of the quadriceps (KF ROM), adductors (HAB-KF ROM), and soleus (ADF-KF ROM) for body balance [52]; in addition, frontal movements may require a wide HE ROM to produce repetitive frontal accelerations [51,52]. These specific technical requirements and movement patterns imposed on BPs are greater in male than in females.

Most authors who have studied the flexibility of BPs have not addressed the analysis of sex differences [47,52,53,54,55,56,57]. Only Hogg, Schmitz, Nguyen and Shultz observed higher values (female 37.9° versus male 28.7°) in internal hip rotation ROM for 50 university BPs in female players [14]. Other studies that investigated ROM in basketball are not comparable to our results because they differ in the method of investigation (starting position, origin of movement (passive vs. active), measurement instrument (goniometer, inclinometer, metric tape measure, Leg-Motion System and units used to measure the data in degrees vs. cm) and sample characteristics such as age, gender, and level of competition [26,47,54,55,56,57,58,59,60,61]. These investigators measured flexibility mainly with linear tests such as the Sit and Reach [54,55,56,57,58] and the Weight-Bearing Lunge [26,47,59,60,61]. The authors also measured the active tests ADF-KE ROM (gastrocnemius) and ADF-KF ROM (soleus) in prone position with a goniometer (angle measurement) to analyze the risk of sports injuries in 42 young university BPs [62]. Notarnicola et al. measured the HF-KE (hamstrings) test (angle measurement) to assess the effectiveness of a stretching program in 30 adolescent BPs [55].

Second, our study demonstrates a comprehensive analysis of flexibility that provides athletic trainers and athletic and conditioning trainers a simple, individualised strategy to improve athletic performance and minimise athletic injuries in players. This comprehensive analysis observed limited ROM with a frequency of 28.1% to 68.8% of BPs for ADF-KF ROM (soleus), KF ROM (quadriceps), HF-KE ROM (hamstrings), HAB-HF ROM (monoarticular adductors), and HE ROM (iliopsoas). The cut-off values for this study are based on reference values from previous studies or clinical experience. In this sense, Backman and Danielson [47] presented a one-year prospective cohort study in which values <36.5° in the ADF-KF ROM were a predisposing factor for patellar tendonitis in young BPs because compensation increases tension on the patellar tendon. Scattone Silva, Nakagawa, Ferreira, García, Santos and Serrão [60] found that a group of senior basketball, handball and volleyball players with patellar tendonitis had 10.7° and 8.9° less in ADF-KF ROM (soleus) and HF-KE ROM (hamstrings) than asymptomatic players, respectively. Cook, Kiss, Khan, Purdam, and Webster [57] found an association between posterior hamstrings extensibility, as measured by the Sit and Reach test, and patellar tendonitis in elite junior BPs. Our individual analysis detected 8 players in the ADF-KF ROM (soleus) and 16 players in the HF-KE ROM (hamstrings) with lower values than the cut-off point proposed by Backman and Danielson [47] and Witvrouw et al. [48]. These players are likely to be at higher risk of future overuse injury if preventive measures are not taken based on the previously cited studies. The excessive training loads (volume and intensity) and repetitive technical skills in basketball lead to physical stress-induced fatigue in the adductors, hamstrings, quadriceps, iliopsoas, and triceps surae of BPs [47,51]. If these changes are not treated with an adequate recovery plan (foam rolling, stretching, or resting), the muscles will limit ROM by decreasing extensibility or tightness of the muscles [63]. In addition, lack of a regular stretching routine could be a major cause of limited ROM and muscle tightness [29,47,52]. Ankle, knee and hip ROM can be improved by a variety of exercises and clinical techniques. For example, the extensibility of the muscles (adductors, hamstrings, quadriceps, iliopsoas, and triceps surae) is a modifiable factor that can serve as a mechanism to mitigate the risk of injury [52,56].

Regarding the inter-limb ROM asymmetry, the results on the magnitude of the standardized mean differences show no inter-limb ROM differences, except for the KF ROM (quadriceps). The mean distance inter-limb ROM in the studied movements was 3° (HE, ADF-KF, HAB, and HAB -HF), 4° (ADF-KE, KF, and HF-KE), and 5° (HF). These results are common in most studies calculating sports inter-limb ROM asymmetry. This result should not be considered valid because the sensitivity of the inclinometer is 2°, and the maximum detectable change (MDC_95_) of the ROM-SPORT tests is between 4° to 7° [38].

However, a comprehensive analysis revealed a frequency of inter-limb ROM asymmetry that ranged from 4.7% to 25% of BPs. The highest frequencies were observed in HE ROM (iliopsoas), ADF-KE ROM (gastrocnemius), KF ROM (quadriceps), ADF-KF ROM (soleus), and HF-KE ROM (hamstrings). Compared to other team sports, the results for the frequency of inter-limb ROM asymmetry are lower than in soccer [15,31], futsal [32], and inline hockey [29], which is due to the fact that the technical actions in basketball are performed symmetrically. This individual analysis revealed 27 players with inter-limb ROM asymmetry of more than 6° (small angles) or 10° (large angles) in one or more ROM movement tests. These players have a higher risk of suffering a sports injury if previous studies are taken as precedent. Fousekis et al. [8] showed that players with KF asymmetry (quadriceps) prospectively suffer from hamstring and quadriceps injuries. Other authors found an association between external hip rotator asymmetry and a clinical history of back pain [34,36,43,45]. Previous studies have reported inter-limb ROM asymmetry in athletes due to unilateral technical movements in sports and greater repetitions and loads with the dominant limb [8,64,65]. The discrepancy in force production and ROM between the dominant and non-dominant limb is caused by functional discrepancies performed with each limb in unilateral dominated sports [66]. The quantitative concept of muscle inter-limb asymmetry (>10% or 15%) reported in the study by Bishop et al. [67] cannot be applied to the tests of ROM-SPORT battery. The main reason is the different angle representing the extensibility of each lower limb muscle. For example, the extensibility of the iliopsoas (HE ROM) was 9° and 15°, and the extensibility of the gluteus maximus (HF ROM) was 145° and 144° in male and female players in this study. The quantitative concept of inter-limb ROM asymmetry of 10% or 15% is different for these muscles (baseline extensibility).

Future research should focus on determining reference values in basketball with a representative sample of players from teams playing in a league. Prospective cohort studies are needed to determine the cut-off value for limited ROM and inter-limb ROM asymmetry for each test to identify BPs most likely to develop pain or injury. In addition, the reference values should be validated with a multifactorial assessment that includes physical and sports performance tests. In this regard, we recommend that researchers focus on the development and evaluation of pre-participation screening and injury prevention programs through high-quality randomised controlled trials targeting athletes at higher risk for basketball injury.

### Practical Application

Athletic trainers and athletic and conditioning trainers have a valuable tool in the ROM-SPORT battery to monitoring the ROM and conditioning in BPs. The normative data presented in this study serve as a baseline for evaluating ROM performance in BPs. A stretching routine could be developed for those players whose values fall below the basketball reference values. The design of this routine should also consider gender differences for athletes in HE, HAB-HF and KF ROMs. A comprehensive analysis may also help athletic trainers and athletic and conditioning trainers to identify musculoskeletal adaptations for sport, e.g., limited ROM by muscle tightness and inter-limb ROM asymmetry in BPs. This individualised ROM analysis is a useful strategy to improve athletic performance and reduce injury risk in competitive basketball players. However, this strategy will always be most effective when a multicomponent training program combining flexibility, strength, and neuromotor is used in basketball players.

## 5. Conclusions

The lower extremity flexibility profiles of the sixty-four basketball players were 15° and 10° HE, 39° and 38° ADF-KE, 40° and 39° ADF-KF, 43° and 43° HAB, 75° and 61° HAB-HF, 78° and 83° HF-KE, 134° and 120° KF, and 145° and 144° HF ROMs of the male and female basketball players, respectively. Comparison of the two lower extremity flexibility profiles shows gender differences for the HE, HAB-HF, and KF ROMs. Basketball players mainly reported muscle tightness in ADF-KF, KF, HAB -HF, HF-KE, HF, and HE ROMs, and inter-limb ROM asymmetry mainly in HE, ADF-KE, KF, ADF-KF, and HF-KE ROMs. This study provides gender-specific lower extremity flexibility profile values in BPs that can help athletic trainers and athletic and conditioning trainers to identify those players who are at higher risk of injury due to abnormal ROM values. Reduction in injury risk can be achieved when professionals use a multicomponent training program that includes flexibility, strength, and neuromotor.

## Figures and Tables

**Figure 1 ijerph-18-11956-f001:**
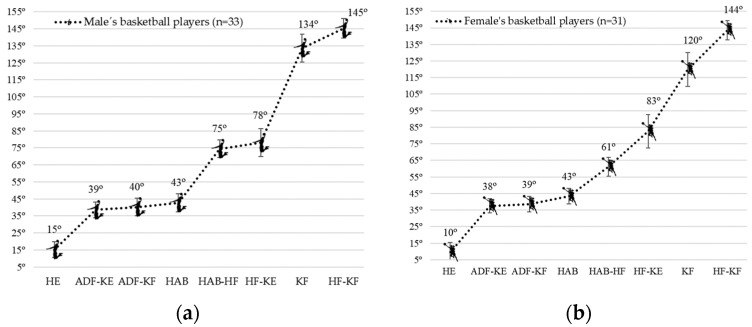
(**a**) Lower extremity flexibility profile for the 33 male basketball players; (**b**) Lower extremity flexibility profile for the 31 female basketball players (mean values of each range of motion test).

**Figure 2 ijerph-18-11956-f002:**
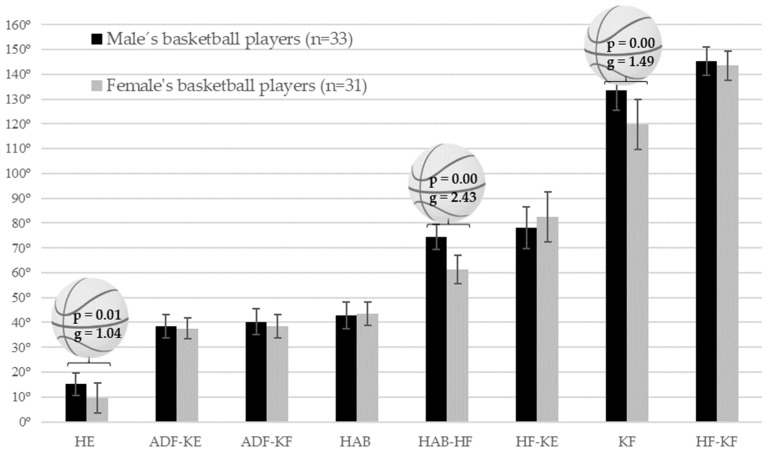
Gender differences in range of motion (mean values of each range of motion test) of the basketball players (*p* = significant differences by gender; g = effect size of the difference).

**Table 1 ijerph-18-11956-t001:** Demographic and sports data from the competitive basketball players.

Variables	Male (n = 33) *	Female (n = 31) *	*p*-Value	ES
Age (years old)	20.6 ± 2.5	24.6 ± 3.2	0.00	−1.38 (Large)
Body mass (kilograms)	89.8 ± 10.9	74.2 ± 11.3	0.00	1.38 (Large)
Height (meters)	193.4 ± 6.2	179.3 ± 7.4	0.00	2.05 (Very large)
BMI (kg/m^2^)	24.1 ± 2.3	23.1 ± 3.1	0.15	0.36 (Small)
Basketball experience (years)	10.2 ± 2.5	14.5 ± 5.3	0.01	−1.04 (Moderate)
Basketball training per week (hours)	8.2 ± 0.4	10.3 ± 0.5	0.00	−4.59 (Extremely large)
Playing time per basketball competition (minutes)	22.3 ± 4.7	19.4 ±6.3	0.04	0.52 (Small)

* Values expressed as mean ± standard deviation; BMI: Body mass index; kg: kilograms; m: meters; ES: Effect sizes Hedges’ g.

**Table 2 ijerph-18-11956-t002:** Range of motion results classified by muscle tightness reference values in 64 competitive basketball players.

Range of Motion Variables	Normal		Tightness		*p*-Value	Effect Sizes Hedges’ g
	Value (°) *	n	Value (°) *	n		
HE (iliopsoas)	16.9 ± 4.2	22	9.2 ± 5.2	42	<0.000	1.56 (Large)
ADF-KE (gastrocnemius)	38.6 ± 4.7	64	-	0	-	-
ADF-KF (soleus)	43.2 ± 4.3	40	29.1 ± 3.4	24	<0.000	3.49 (Very large)
HAB (adductors)	42.8 ± 5.4	64	-	-	-	-
HAB-HF (monarticular adductors)	81.1 ± 4.6	10	67.2 ± 6.3	44	<0.000	2.27 (Large)
HF-KE (hamstrings)	93.3 ± 7.8	25	73.4 ± 8.4	39	<0.000	2.41 (Very large)
KF (quadriceps)	139.2 ± 7.7	27	121.9 ± 9.2	37	<0.000	1.99 (Large)
HF (gluteus maximus)	145.9 ± 5.9	46	134.7 ± 4.9	18	<0.000	1.96 (Large)

* Values expressed in degree as mean ± standard deviation for each group (tightness versus normal); HE: hip extension; ADF-KE: dorsiflexion of ankle with knee extended at 0°; ADF-KF: dorsiflexion of ankle with knee flexed; HAB: hip abduction with knee extended at 0°; HAB-HF: hip abduction with hip flexed; HF-KE: Hip flexion with knee extended at 0°; KF: knee flexion test.

**Table 3 ijerph-18-11956-t003:** Inter-limb ROM asymmetries (favoring dominant or non-dominant limb) in each movement assessed of basketball players.

Range of Motion Variables	Total (%)	Dominant Limb (n)	Non-Dominant Limb (n)
HE (iliopsoas)	16 (25.0%)	9	7
ADF-KE (gastrocnemius)	14 (21.9%)	3	11
ADF-KF (soleus)	12 (18.8%)	4	8
HAB (adductors)	11 (17.2%)	5	6
HAB-HF (monoarticular adductors)	3 (4.7%)	1	2
HF-KE (hamstrings)	11 (17.2%)	4	7
KF (quadriceps)	14 (21.9%)	8	6
HF (gluteus maximus)	5 (7.8%)	3	2
Total sample	86 (100%)	37	49

n: Value represent number of basketball players; HE: hip extension; ADF-KE: dorsiflexion of ankle with knee extended at 0°; ADF-KF: dorsiflexion of ankle with knee flexed; HAB: hip abduction with knee extended at 0°; HAB-HF: hip abduction with hip flexed; HF-KE: Hip flexion with knee extended at 0°; KF: knee flexion test.

## Data Availability

The data sets used and analyzed during the current study are available from the first or corresponding author on reasonable request.

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
