# Peer review of "Lower Extremity Flexibility Profile in Basketball Players: Gender Differences and Injury Risk Identification"

_ijerph, 2021, doi:10.3390/ijerph182211956_

Round 1
Reviewer 1 Report
Reviewer Report
Lower extremity flexibility profile in basketball players: gender differences and injury risk identification
I think this study will provide efficient value for the prevention of sports injuries.
[Introduction]
Ln61: Lower extremity flexibility profile values are determined by performing tests that integrate the standardised ROM-SPORT battery for a group of athletes.
# To be considerate of the reader, please explain the concept of ROM-SPORT battery in an easy-to-understand manner. I would like you to explain it in detail in the Research Methods section.
[Methods]
Ln112: optimal cut-off values for predicting injury risk
# The description of the reference values above is lacking. The cut-off values are relevant to the validity of this study result, so please add a detailed explanation to the study method part.
Ln116: 17 and 31 years (21.9 ± 3.9 years)
# You explained in the introduction that ROM is age-dependent. The age range of the subjects in this study is 17-31 years old, so it is large. Therefore, I think the extensive age range of the subjects in this study will affect the ROM measurement results. Add your comments on this issue to the discussion section.
Ln116-117: (21.9 ± 3.9 years), (12.8 ± 4.6 years)
# I wonder if the age and career values of the subjects you suggested are the average of the 64 subjects in this study. Because when I calculated the pooled average using the average value and sample size you suggested in Table 1, it is different from the overall average you told. I hope you explain this part correctly.
Ln139-140: self-reported age, basic anthropometric data (age, height,...)
# You double-presented 'age'. Delete one.
Ln179-182: A sample size of 33 female and 31 male basketball players, an alpha level of p < 0.05, effect size (Hedges' g), and Student's t-test for independent samples (Figure 2) were used for power of sample size analyses.
# The explanation of the sample size calculation basis is very insufficient. Would you please explain in detail?
Ln183, 185:
# Paired two-sample t-test => Paired sample t-test
[Results]
Ln198: Nine players were withdrawn from the study by the researchers.
# Does your explanation mean that 9 out of 64 were excluded, or does it mean that 9 out of 75 were excluded? Would you please explain exactly?
Ln203: 0.99 for EC, ...
# The full name of 'EC' does not exist. Would you please present the full name of 'EC'?
Ln 208:
# There is a lack of explanation for the results in Figure 1. Add a brief interpretation of what the angles you give in Figure 1 mean.
# Overall, there is a lack of explanation of what the results mean. If you add an easy explanation of the results, this paper will be a better one to read.
[Conclusions]
Ln352: twenty-four basketball players were found to be 15° and 10° HE, ...
# Make sure twenty-four is correct.
Ln358-361: This study provides gender-specific lower extremity flexibility profile values in BPs that can help athletic trainers and athletic & conditioning trainers to identify those players who are at higher risk of injury due to abnormal ROM values
# A period is missing at the end of this sentence.
Author Response
Dear Reviewer,
Thank you for your review and constructive comments. We have reviewed the manuscript and addressed all your remarks (see the responses below). We hope that you consider the reviewed version of the manuscript worthy for publication.
Kind regards,
The author.

Reviewer 2 Report
General Comments to Author(s):
The authors have done a good job at addressing why establishing a flexibility profile for basketball players is important and have designed a study to help address this issue. However, the manuscript contains various issues that need further attention. See below for specific comments.
Specific Comments
Introduction
- The authors have done a good job explaining range of motion, how it is measured, why it is important for athletic performance, and why the current study is needed.
Methods
- 1. Design of study: Consider adding a brief description of why participants were instructed to participate in no vigorous exercise for 24 hours before testing. It could be argued that a longer time would be needed to ensure DOMS would not be a potential limiting factor in determining participants ROM.
- 2: Pertaining to the above comment, it is stated that players experiencing DOMS were omitted from the study. How was it determined if participants were experiencing DOMS? What was the criteria for excluding participants from the study?
- 5: Why was only one knee measured? (may be a typo)
Results
- 1st paragraph: Reword to avoid using personal pronouns (they).
- Table 2: Please add the units that Normal and Tightness values are expressed in.
- Table 3: Check formatting to ensure that all of the table is on the same page.
Discussion and Conclusion
- The discussion is thorough and utilizes comparisons to previous research when describing the interpretation of the results and points out future potential paths to build off of the current study.
- As stated above, check throughout paper for uses of personal pronouns and reword when needed to remove those uses.
Author Response

(The authors gave the same response as above.)

Reviewer 3 Report
Thank you for asking me to review the current work entitled ‘Lower Extremity Flexibility Profile in Basketball Players: Gender Differences and Injury Risk Identification. This is a well written piece of work. That said, I do feel there needs to be more balance in relation to the narrative surrounding injury risk. The paper is suggestive that this is a one size fits all approach and flexibility cures all. There needs to be an acknowledgement and discussion of the findings in the present study in terms of the multi factorial nature of performance and injury risk.
Introduction:
Line 36 -38 – the authors discuss the main objectives to measuring ROM. I would argue current literature only supports this if flexibility is increased relative to functional strength. There have been several studies completed identifying that without this approach the increases in flexibility could expose them to increased risk of injury due to causing a NM delay when performing functionally. I would suggest reviewing this.
Line 42-55: is there a difference between ROM and functional ROM – i.e. what ROM the athlete needs specific to their sports performance. I think discussion is needed here for clarity. Generally, it there is to much emphasis on flexibility as a sole factor to reduce injury/protect the athlete and increase performance. I think the discussion within the intro requires more balance within the narrative. We need to be clear where we are now in terms of flexibility, the importance of it in relation to other factors that are key to increasing performance and decreasing injury risk, and where we need to go to. This will provide the intro with more balance, but reasonably justify the need for the work. Also, there is limited discussion with regards gender differences and effects these have on ROM measures.
Materials and Methods:
Line 103: Define experienced MSK experts – how experienced, who are they etc.
Line 104: How was this randomised and exactly what?
Line 105: Why was this set at a 5% difference, what is the justification for this? Why was average of two measures taken and not the 3? Why was best score not used? Provide more detailed justification.
Line 109-110: Why was there a pause and why these time frames?
Line 139: was anthropometric data not taken by the researchers and reliant on the survey? If so this would reduce the reliability of the information detailed with regards the physical characteristics of the subjects.
Results:
Line 201: You do not detail in the methods the exclusion criteria in relation to injury. Please do this.
Discussion:
Line 248: Why is this surprising?
Line 240-262 – I enjoyed reading the summary of the main findings and with interest male and female differences. You site movement exposure as a key component to suggest why this may be the case. Contradicting previous literature and findings. However, this goes back to the points made surrounding the intro. To reduce injury risk and increase performance, flexibility alongside increased functional strength, power etc will increase performance and reduce injury risk, not flexibility alone. Points raised in both the intro and discussion must reflect this.
Line 278: What is this individualised strategy in relation to flexibility that increases performance?
Line 289-294: To draw these assumptions you would need to site RCT research or systematic reviews to conclude there is a link between flexibility and overuse injury. Functional range not more important? – this is the whole discussion surrounding functional range and ankle stiffness to perform sprinting effectively.
Line 334: Do you not think future research should focus on multi factorial measures related to performance?
Conclusion:
Line 360: I agree it may contribute to reducing injury risk, but is not the only factor and the authors must reflect this in the conclusions drawn.
Author Response
Dear Reviewer,
Thank you for your review and constructive comments. We have reviewed the manuscript and addressed all your remarks (see the responses below). We hope that you consider the reviewed version of the manuscript worthy for publication.
Kind regards,
The authors.

Reviewer 4 Report
First of all, I would like to congratulate the authors for their study, the theme and the great meticulousness with which they prepare the work.
As the authors say, in the field of sports and health sciences, measuring ROM is important to identify flexibility-related risk factors such as muscle tightness and asymmetries for athletic performance and sports injuries, and 2) because it is necessary to evaluate the physical-technical training process and rehabilitation of sports injuries.
On the other hand, they present a representative sample in order to offer interesting results.
The methodology is very clear and describes the entire process very well. Furthermore, the authors use valid and reliable instruments.
As possible improvements it could be said that the number 0.000 in science does not exist and perhaps it would be better to use the <0.000
And on the other hand, comment that the tables and titles should be prevented from being cut.
In short, it has been a pleasure to read your article and learn from your work.
Thank you very much
Author Response

(The authors gave the same response as above.)

Round 2
Reviewer 2 Report
The authors have addressed my previous comments and concerns. I believe the paper has been improved. See below for specific comments.
Specific Comments
Methods, 2.1: Consider adding an explanation for why a time frame of 24 hours of no intense exercise was chosen as opposed to 48 hours as previous research has shown that DOMS peaks at 48 hours post exercise.
Author Response
Dear reviewer,
We have reviewed the database and the participants did not perform vigorous physical exercise 48 hours before the evaluation session. The data has been modified in the manuscript.
English language and style will be reviewed in the proof.
I appreciate the contributions to improve the manuscript.
Best regards,
The author

Reviewer 3 Report
Thank you for the responses to the comments made - i would recommend this paper for publication
Author Response
Dear Reviewer,
English language and style will be reviewed in the proof.
I appreciate your contributions to improve the manuscript!
Antonio Cejudo